# Influence of the Acoustic Cover of the Modular Expansion Joint on the Acoustic Climate in the Bridge Structure Surroundings

**DOI:** 10.3390/ma13122842

**Published:** 2020-06-25

**Authors:** Janusz Bohatkiewicz, Michał Jukowski, Maciej Hałucha, Marcin Dębiński

**Affiliations:** 1Faculty of Civil Engineering and Architecture, Lublin University of Technology, 40 Nadbystrzycka Str., 20-618 Lublin, Poland; m.jukowski@pollub.pl (M.J.); m.debinski@pollub.pl (M.D.); 2EKKOM Sp. z o. o., 71B dr. Józefa Babińskiego Str., 30-394 Krakow, Poland; m.halucha@pollub.pl

**Keywords:** bridge expansion joint, impulse noise, acoustic cover

## Abstract

The noise generated at the interface between the wheels of vehicles and the road surface is well recognized in the literature worldwide. Many publications describe the phenomenon of reducing this kind of impact by silent road surfaces. A specific type of this noise is the sound generated by vehicles passing over the expansion joints of bridge structures. Due to the impulsive nature of this sound, it is very onerous for people living in the close vicinity of bridge structures. The passage of vehicles over expansion joints causes the formation of vibrations that are transmitted to the structural elements of bridge structures, which may cause the formation of the material sounds (especially arduous in the case of bridges with steel elements). An attempt to reduce this impact was made by making a prototype acoustic cover of the expansion joint on the selected bridge. The paper presents the results of research on the “in situ” acoustic effectiveness of this cover. Additionally, the noise was modelled in the object surroundings before and after the cover’s application. The acoustic efficiency of the cover in the whole measured frequency range was 5.3 dBA. In the narrower frequency bands (1/3 octave bands), larger sound level reductions were observed. The maximum sound levels measured under the tested dilatation were less than 10.0 dBA lower than the maximum sound levels measured under the reference dilatation.

## 1. Introduction

A bridge expansion joint is a structural component designed to provide smooth passage over a gap between adjacent sides of a bridge deck. This gap is made in a purposeful way to compensate deformations in the bridge construction elements. These deformations may be caused by: time-varying loads, temperature amplitude, rheological processes of the structural elements, or compression of the object. The selection of an expansion joint is primarily dictated by the nominal values of the displacements for which a particular type of expansion joint is designed. The widths of the dilatation gap may vary from a few to several tens of centimeters in the case of standard types of objects, and in the case of suspended or hanging structures, they may be larger than 1 m [1]. The basis for dimensioning the width of dilatation devices is the value of displacement along the object axis. It influences the noise level generated by vehicles passing over a bridge. Particularly high nuisance is caused by lorries and those vehicles moving at high speeds.

The problem of the acoustic impact of bridge expansion joints is a global problem. Many publications have been created describing the analyzed problem and solutions in individual countries, e.g., in Europe [2,3], Japan [4], China [5], and Australia [6,7]. This noise is connected to the vehicle passing over the expansion joint, the so called “joint sound”, and the vehicle passing over the object, the so called “span sound”. Issues related to this type of noise are assigned to the environmental noise issues. The essence of the “span sound” has been described in detail, among others, in the papers [8,9]. The phenomenon of low-frequency noise is closely related to the propagation of vibrations in the structural elements of bridges. It has been proven that the level of low-frequency noise and the level of vibration amplitude of bridge construction elements is mainly influenced by the speed of vehicles and the type of suspension. The pneumatic suspension vehicles have shown less vibration amplitude than the leaf-type suspension vehicles. [10].

The phenomenon of the acoustic impact generated by vehicles passing over expansion joints also applies to railway bridges, where there are the much higher speeds of trains in comparison to road facilities. This has been examined and described in papers [5,11]. In such cases, the material from which the bridge structure was built is of key importance. In the case of steel bridges, the noise level from the passage of a train is usually about 10 dB higher than in the case of a concrete bridge [12].

The modular expansion joints are most often used in bridge structures. They are characterized by high watertightness, geometric adaptation, and the ability for progressive and rotary movement. Unfortunately, from an acoustic point of view, they do not obtain such good results. Despite the use of all types of silencing covers (sinusoidal or rhombus steel plates) on the top layer of the expansion joint, due to cyclic, dynamic loads on vehicle wheels, they are destroyed and no positive acoustic effect is obtained from their use [13,14,15,16].

Due to the number of advantages, modular devices have been tested for a deeper understanding of the noise generation mechanism. It was found that noise generated above the device is dominated by frequency components in the range of 500–800 Hz [1], and the noise generated below it is dominated by the frequency components in the range up to 200 Hz [1,4]. The value of the noise level under the device may be affected by the acoustic characteristics of the space below the expansion joint. In addition, resonance may occur—the dominant frequencies of the construction elements (beams) of the device may coincide with the frequencies of the sound level below the expansion joint.

The road noise and impulse noise generated at the junction between the bridge expansion joints and vehicle tires, in addition to having a negative impact on the local community, also have a negative impact on the use of the passageways by wild animals. The noise generated by wheels on the dilatation increases the impact even by 12 dB and may cause deterrence of animals (mainly mammals), causing unsuccessful migration [3]. This study shows that from the point of view of migration of wild animals, passages without expansion joints are the most beneficial.

The authors of this paper have attempted to assess the impact of the application of an innovative solution, i.e., an acoustic cover, mounted directly under the modular expansion joints, on the acoustic climate in the surroundings of the selected bridge facility. A similar solution, but at the same time different in terms of construction and application possibilities, is presented in the papers [17,18]. In this case, the application of the so called Helmholtz resonator was described, which on the Anzac and Georges River (Tom Ugly) bridges resulted in the reduction in the noise defined as the “booming noise” by as much as 10 dB in the frequency band up to 200 Hz. Another possible solution is the so called ROBO MUTE system, designed to close the space directly under the expansion joint. According to reference [19], this system allows noise reduction up to 19 dB. Unfortunately, no more detailed information on the parameters was specified, e.g., frequency, in which such an acoustic effect was obtained. It resembles closer the innovative solution of the acoustic cover described by the authors of the paper. In order to obtain a better effect of the impulse noise reduction on the modular expansion joints, together with the ROBO MUTE system, it is possible to additionally use the ROBO FOAM system. This enables a reduction in the impulse noise caused by the compression of air as a wheel crosses a gap, by filling the space between the bars of the modular device with a special rubber foam. A highly similar solution to the ROBO MUTE system is a flexible anti-noise system mounted under the expansion joint structure developed by the MAURER SÖHNE, the MAURER GU-f model [20]. The solution is to suspend a lightweight sound-absorbing material under expansion joints and connect it in a flexible way to the object. This solution allows easy access to the expansion joint from below, because it is quickly disassembled by one person. Measurements carried out by the manufacturer have shown that this solution is very effective and allows for the reduction in noise measured directly at the expansion joint by more than 15 dBA. 

## 2. Characteristics of the Tested Bridge Structure and the Expansion Joints Used 

The bridge structure, in which the expansion joint selected for the research was used, is located in the Silesian Voivodeship in southern Poland. It leads vehicle traffic on the A1 motorway—one of the main national roads. It is a multi-span structure, consisting of 15 free-spans and 4 continuous spans. The supporting structure was made from combined technology in the form of steel girders and crossbars connected to the reinforced concrete slab of the platform. Modular expansion joints were used in the bridge (Figure 1). From a technical point of view, these devices were designed and made correctly. 

The dominant acoustic effect occurring in the facility surroundings is the noise of the impulse nature, generated by the vehicles passing over the expansion joints. This impact is minimized only from the roadway side through the acoustic screens. It is not limited from the bottom of the object. The structure of the bridge also has a significant impact on the propagation of noise to the environment. Vibrations generated when a vehicle passes through the expansion joints generate the material sound in steel elements. The pond located underneath the object additionally influences the worsening of acoustic conditions, as the sound is reflected from the water surface (this causes its amplification, because the reflected wave adds to the incident wave). A lesser impact on the acoustic climate has a typical acoustic impact generated by passing vehicles—the rolling noise and engine/transmission noise. It is minimized by noise screens. The acoustic conditions in the vicinity of the tested object have a negative impact on human health (mainly due to the sleep disorders—described in detail, among others, in reference [21]).

## 3. Research Methodology

The methodology to perform the “in situ” tests was developed on the basis of Annex 11 in the report [22] and it was modified for the assumed purpose: the determination of the acoustic efficiency of the expansion joint cover. The noise level measurements were made using the sound level meters with 1/3 octave filters. For measurements, a real sound source was used, which was the noise generated by vehicles passing over the expansion joints. From the point of view of the purpose of the measurements, the use of a real sound source was justified because the acoustic cover should be tested in such conditions that will actually occur on the bridge structure in question.

The “in situ” measurement results were expressed using the equivalent sound level. This indicator was selected for analysis due to the purpose of using the acoustic cover, which was the minimizing of the acoustic impact of the expansion joints in the vicinity of the bridge (the noise generated by vehicles traveling over expansion joints had to be compared with the permissible values, which in Poland, are expressed using the equivalent A-weighted sound level). 

In the first step, tests were carried out at a point located at a height of about 1.5 m above the tier level of the object support, under the tested expansion joint. Based on the results, it was possible to determine the frequency characteristics of the sound generated under the tested dilatation. It was used to select the acoustic parameters of materials (insulating and absorbing) that were used to make the cover. Then, tests were carried out under the expansion joint device selected for noise protection and under a reference dilatation (located under an adjacent support), to which the test results were compared. These tests were carried out both before and after applying the acoustic cover on the tested expansion joint. The results of these tests were then used to calculate the acoustic efficiency of the cover.

There was a very high level of an acoustic background in the vicinity of the examined bridge. For this reason, it was decided to locate the measuring points directly under the tested expansion joints. Only in such a location was it possible to perform tests, the results of which could be used to determine the acoustic efficiency of the expansion joint cover. This location also ensured the equivalence of field conditions, disturbing obstacles and reflecting surfaces (the construction of caps, girders and other elements of the object was very similar under the tested and reference dilatation). The location of the measurement points (before and after applying the cover) is shown in Figure 2.

On the basis of the test results, the acoustic efficiency and frequency characteristics of the noise protection were determined. For this purpose, the following relationship was used:(1)$$D_{IL}=\left(L_{bad,A}-L_{bad,B}\right)-\left(L_{ref,A}-L_{ref,B}\right)$$
where: 

*D_IL_*—the “in situ” sound protection efficiency (dBA);

*L_bad,A_*—the equivalent sound level measured under the tested dilatation (without the use of the cover) (dBA);

*L_bad,B_*—the equivalent sound level measured under the tested dilatation (with the cover applied) (dBA);

*L_ref,A_*—the equivalent sound level measured under the reference dilatation (the situation without the cover used on the tested dilatation) (dBA);

*L_ref,B_*—the equivalent sound level measured under the reference dilatation (the situation with the cover used on the tested dilatation) (dBA).

Due to the fact that the acoustic measurements for the tested and reference expansion joints were carried out in parallel (at the same time), and these devices were located on the same highway road at a distance of about 40 m (Figure 3), it should be assumed that traffic conditions (traffic volume, vehicles speed, percentage of heavy vehicles) did not affect the test results. The only parameter that could affect the discrepancy in the equivalence of the sound source for the tested and reference dilatations could have been the design of the expansion joint device itself, its technical condition and the difference in the sound level generated by passing vehicles through these devices. However, these differences were taken into account in the calculations of the acoustic efficiency of the cover by including the results of measurements made at both expansion joints after dismantling the cover, in accordance with the relationship presented above.

The acoustic tests were additionally carried out at points located next to the bridge structure. However, in these places, the acoustic noise generated by other expansion joints was so great that they prevented obtaining reliable data for the analysis. These results were not presented later in the paper. On their basis, it can be concluded that the improvement of acoustic conditions in the vicinity of the bridge structure will occur when noise protection is used on all expansion joints. Due to the fact that it was not possible to apply at the pilot testing stage, the acoustic modeling was performed, which included noise reduction after the application of covers on all expansion joints. For this purpose, the French calculation method NMPB Routes-96 (Guide du Bruit) was used. The actual traffic volume (51,946 vehicles per day) and vehicle speed (120 km/h for light vehicles and 85 km/h for heavy vehicles) measured during the measurements were used as input. The type of vehicle was also taken into account (22% heavy vehicles). SoundPLAN software (version: 8.0, SoundPLAN GmbH, Backnang, Germany) was used for the acoustic calculations. The view of the geometric model, in which all elements relevant to the sound emission and propagation are mapped, is presented in Figure 4.

Due to the lack of mapping the noise generated by the expansion joints in the available calculation models, it was decided to use linear sources of industrial noise for this purpose. For the acoustic calculations of industrial noise, the method described in “ISO 9613-2: 1996 Acoustics—Attenuation of sound during propagation outdoors—Part 2: General method of calculation” was used. The acoustic model was then verified with the results of the “in situ” measurements. The results of the model were consistent with the results of the acoustic measurements taken at the same points. Thanks to this, it was possible to determine the acoustic conditions in the vicinity of the bridge after the application of covers on all expansion joints. 

The results of the acoustic tests and conclusions formulated in this regard are presented in detail in Section 4 of the study.

## 4. Description of the Noise Protection Applied

In order to determine the reduction in noise generated by vehicles passing over the expansion joints of the tested object, a prototype constructional solution of the sound-absorbing and insulating cover (hereinafter also referred to as “the sound cover”) was developed. The main components adopted at the initial concept stage, apart from the need to ensure appropriate reduction in the acoustic impact, were the ease of assembly and disassembly of this device. The construction of the cover had to ensure the possibility of easy access to the expansion joints by the motorway maintenance services.

The most important elements of the acoustic cover were the sound insulating panels and sound absorbing material fixed to the scaffolding structure. It should be stressed that the target solution for the construction of the acoustic cover must ensure the possibility of changing its width resulting from the movement of the bridge decks and the operation of the expansion joints. Due to the short time performing the acoustic tests (several hours), the prototype of the acoustic cover was equipped with a structure consisting of supports (flat bars) of fixed length, which did not allow changing of the width of the device. The technical drawing showing the bottom view of the expansion joint for the two modules (mounting units) of the sound cover is shown in Figure 5.

The elements of the cover that determined its acoustic effectiveness were sound insulating boards and sound absorbing materials. First of all, a material was chosen to isolate the noise caused by vehicles passing over the dilatation. For this purpose, 4 mm thick solid polycarbonate slabs were selected and used, which had the weighted sound reduction index *R_w_* = 27 dB.

Each assembly module consisted of two plates with widths of 50 and 75 cm. They were mounted to the supporting structures (flat bars), which made their longitudinal movement (resulting from the object transverse inclination) impossible. The two plates overlapped each other, creating an overlap to enable the change of width of the cover caused by the longitudinal movement of the platform slabs. This overlap also minimized the decrease in the “in situ” acoustic efficiency of the cover (an effect similar to the one observed when connecting two acoustic shields). The method of assembly of polycarbonate slabs to the supporting structures using the overlap is presented below in Figure 6.

The second element, important from the point of view of noise reduction by the sound cover, was the sound absorbing material. Its task was to absorb the sound generated by vehicles passing over the tested dilatation, which limited the influence of reflected sounds inside the cover from the structure of the bridge, dilatation and surface of the cover. In order to properly select the parameters of this material, preliminary acoustic measurements were made (the methodology of performing these tests is presented in Section 3). These measurements were important due to the fact that materials of this type effectively absorb sound only in a narrow frequency range, with this range being different for each type of material (it also depends on its thickness). Improper selection of parameters may result in insufficient noise absorption and an increase in the sound level inside the cover, which may affect the acoustic effectiveness of the cover. The results of these measurements are shown below in Figure 7.

The dominant noise levels (higher than 70 dBA) are found in the 1/3 octave band middle frequencies in the range from 125 to 1000 Hz. The materials used to fill the interior of the cover should therefore absorb sound primarily at these frequencies. It was therefore decided to use a combination of two types of materials with different frequency characteristics—the acoustic foam. The first one was a “pyramid-shaped” material, which could easily be used to fill the interior of the cover (the average sound absorption coefficient—0.29). This material has higher values of the sound absorption coefficients for higher frequencies. The second was a material called as the “bass traps” (the average sound absorption coefficient—0.72). Information on the sound absorption curves for the materials used is shown in Figure 8. On the other hand, it was characterized by higher values of the sound absorption coefficient for lower frequencies. A photo showing the used sound absorbing materials from the inside of the sound cover is shown in Figure 9.

From an acoustic point of view, the “tightness” of the acoustic cover was very important. Each gap could cause a decrease in the effectiveness of the device and lead to a situation in which most of the acoustic energy would be generated to the environment. In order to seal discontinuities (gaps) formed at the contact between the solid polycarbonate slab and the vertical surface of the concrete, the silicone was used at all edges of the cover (Figure 10).

The results of the in situ acoustic efficiency of the cover are presented in the next section of the paper.

## 5. Results of Measurements

The results of measurements in the form of the equivalent sound level (measured over the whole observed range) as well as sound levels in the middle frequencies of 1/3 octave bands are shown below in Figure 11. The graph shows the results of measurements made at the same time at points located under the tested dilatation with the applied acoustic cover and under the reference dilatation.

The equivalent sound level measured under the tested dilatation for which the acoustic cover was used is 7.0 dBA lower than the equivalent sound level measured under the reference dilatation. The prototype acoustic cover has resulted in a significant reduction in the acoustic impact of vehicles passing over the tested expansion joint compared to the reference one. The decrease in the sound level is visible in each of the 1/3 octave bands. This proves the correct selection of sound insulating materials and parameters of the absorbing materials; they have sufficiently reduced the noise generated inside the sound cover.

Noise reduction under the tested expansion joint occurred despite the fact that the influence of material sound generated by the steel structure of the object (induced by vibrations of the expansion joint elements) was not limited in its vicinity. This noise interfered with the results of the proper measurement, although its level is much lower than the impulsive sound generated by the vehicle’s passages.

In order to calculate the “in situ” acoustic performance of the expansion joint cover, it was necessary to carry out additional comparative measurements of the tested and reference expansion joint in the absence of the acoustic cover. Such measurements were made at the same measuring points after dismantling the sound cover. The results are presented below in Figure 12.

The results of the comparative acoustic measurements show slight differences in the noise generated by both expansion joint devices. It should be noted that the tested expansion joint was about 1.7 dBA quieter than the reference one (taking into account the results of measurements of the equivalent sound level in the whole observed range). These differences were also visible in the case of noise levels measured at individual 1/3 octave band middle frequencies. These data were used to calculate the “in situ” acoustic effectiveness of the cover, which was performed according to the relation described in Section 3. The results of these calculations are shown below in Figure 13. They show the actual effectiveness of the acoustic cover, taking into account the correction coefficients resulting from the differences in the acoustic impact generated by both dilatations in the absence of the cover on one of them.

The sound cover in the “in situ” conditions in which the measurements were made resulted in a reduction in the equivalent sound level of approximately 5.3 dBA over the whole frequency range being measured. A reduction in the noise level could also be observed in each 1/3 octave band middle frequency. The differences are the greatest for the frequency range from 160 to 630 Hz (in each case, greater than 5.0 dBA). This indicates correct selection of the parameters of sound absorbing materials, which were characterized by the highest values of the sound absorption coefficient in this range. A high acoustic efficiency for high frequencies (higher than 4 kHz) can also be observed. In this case, it results from the fact that these frequencies were more effectively isolated by the polycarbonate board (a sound insulating material in the sound cover). The effectiveness of the sound cover under the “in situ” conditions should be described as very good.

The main purpose of the use of the acoustic cover is to reduce the impulse noise generated by vehicles passing over the expansion joints. This phenomenon is characterized by high values of the sound level lasting a short time. Therefore, the acoustic cover is primarily designed to lower the maximum sound levels (the so called “peaks” or “booming noise”). The maximum sound levels measured under the expansion joint with the sound cover are over 10.0 dBA lower than the maximum sound levels measured under the reference joint. It should be stressed, however, that the cover did not reduce material noise generated in the steel structure of the object. However, the level of this impact is much lower than the impulse noise generated by the expansion joints.

The results of the “in situ” measurements made it possible to determine the acoustic effectiveness of the acoustic cover applied on the one selected expansion joint. In order to determine the extent to which the acoustic conditions in the vicinity of the bridge will improve after the application of the cover on all expansion joints, noise modelling was performed. The results of these calculations (in the form of contour lines with acceptable sound levels for the day and night time) are shown in Figure 14 and Figure 15.

The use of the acoustic covers on all dilatation devices of the bridge structure will improve the condition of the acoustic climate in its surroundings. It should be emphasized, however, that first of all, the impulse noise will be reduced, which cannot be directly observed on the basis of the results of calculations expressed using the equivalent sound level. This is due to the nature of the impulse sound, which affects in a very short time and does not have a significant impact on the equivalent sound level (averaged during the day or night time). The use of the acoustic covers will reduce the impulse noise, which is a significant nuisance for people living in the vicinity of the bridge.

## 6. Discussion

This paper presents the results of the “in situ” research on the acoustic effectiveness of the pilot acoustic cover of a selected expansion joint device. In the whole measured frequency range, this efficiency was equal to 5.3 dBA. In the narrower frequency range of the 1/3 octave bands, larger sound level reductions were observed. For the frequency range from 160 to 630 Hz, they were not less than 5.0 dBA, which indicated the correct selection of parameters of the sound absorbing materials. The maximum sound levels measured under the tested expansion joint were less than 10.0 dBA lower than the maximum sound levels measured under the reference expansion joint. This demonstrates the high effectiveness of the prototype acoustic cover from the point of view of the goal to be achieved—reducing the level of the impulse noise generated by vehicles passing over expansion joints. The reduction in the maximum sound level of this nature has been achieved to a great extent. These devices will not reduce the material noise generated in the steel structure of the facility. However, the level of this impact is definitely lower than the noise generated by the expansion joints. Comparing the obtained results of noise reduction to other available systems described in the first section of the paper, it should be emphasized that there is no uniform methodology for measuring the acoustic efficiency of devices of this type. Therefore, the test results may differ significantly not only because of the acoustic properties of the devices, but also because of the different way of measuring them. The use of the Helmholtz Resonator [17,18] reduced the sound level by as much as 10 dB, but only in the specified frequency band (up to 200 Hz). Based on the conducted tests of the analyzed acoustic cover, it was found that for the 160 and 200 Hz bands, a sound level reduction of approximately 7 dBA was obtained. Manufacturers of available systems [19,20] present a noise reduction value of approximately 15 dBA expressed as the maximum A weighted sound level. Therefore, taking into account the above, it should be stated that the tested solution has a slightly lower, but comparable, acoustic efficiency to other available devices on the market. However, its lower cost is a big advantage. 

The results of the acoustic modeling showed that the use of the acoustic covers on all expansion joints of the bridge structure will improve the condition of the acoustic climate in its surroundings. It should be emphasized, however, that their use will above all reduce the impulse noise, which cannot be observed on the basis of the calculation results using the equivalent sound level. This is due to the nature of the impulse sound, which affects in a very short time and does not have a significant impact on the equivalent sound level (averaged during the day or night time). The research, the results of which are presented in this paper, were performed only on one selected bridge expansion joint. They should be continued after applying the acoustic covers on all expansion joints. This is a direction of further research that will determine whether the use of the acoustic cover at all expansion joints can significantly reduce the onerous impact in the surroundings of the object. From an acoustic point of view, it would be necessary to verify the use of other materials for the construction of the acoustic cover, e.g., glass wool, and to make a detailed analysis of their impact on noise reduction.

## Figures and Tables

**Figure 1 materials-13-02842-f001:**
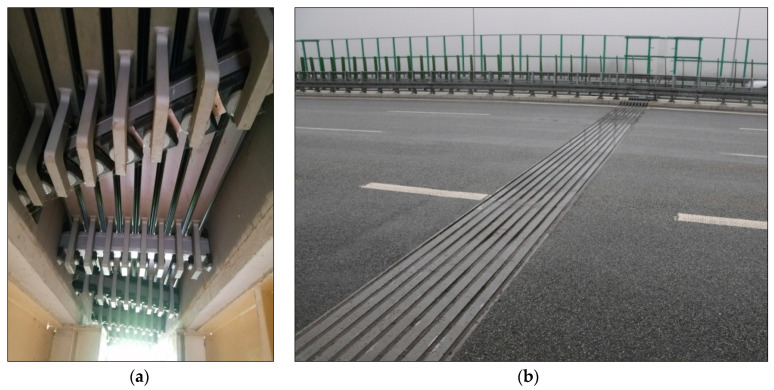
The modular expansion joints applied in the bridge: (**a**) view of the tested expansion joint from the bottom of the object; (**b**) view of the tested expansion joint from the road surface side.

**Figure 2 materials-13-02842-f002:**
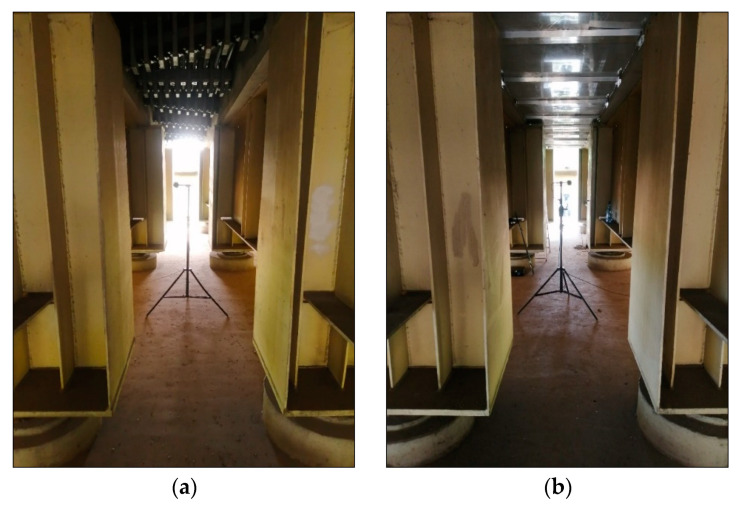
The location of: (**a**) the measurement microphone under the expansion joint device without the cover used (the reference expansion joint); (**b**) the measurement microphone under the expansion joint device with the cover used (the tested dilatation).

**Figure 3 materials-13-02842-f003:**
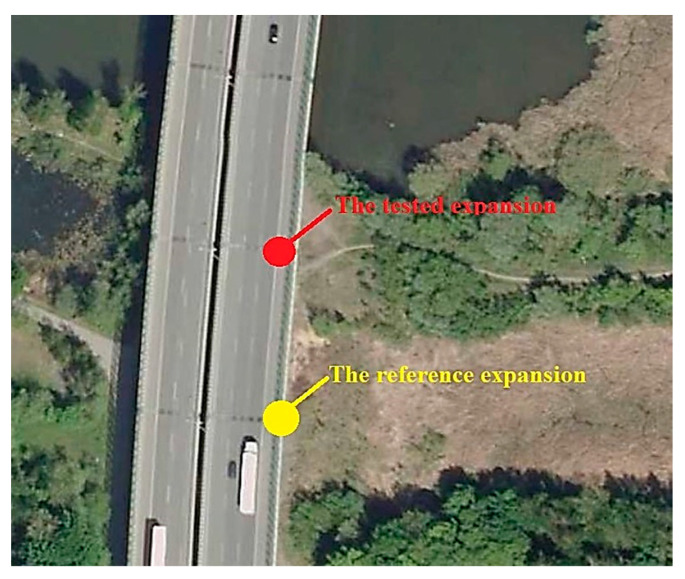
Location of the tested and reference expansion joint.

**Figure 4 materials-13-02842-f004:**
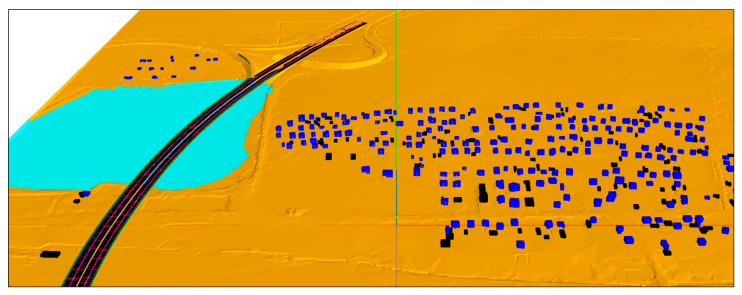
View of the geometric model used for the acoustic modeling.

**Figure 5 materials-13-02842-f005:**
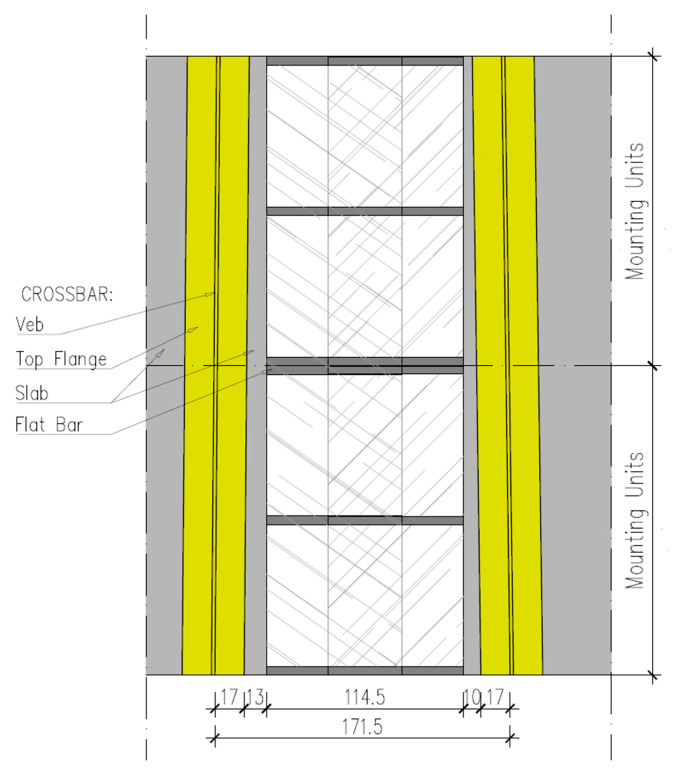
View from the underside of the expansion joint on two modules (mounting units) of the sound cover.

**Figure 6 materials-13-02842-f006:**
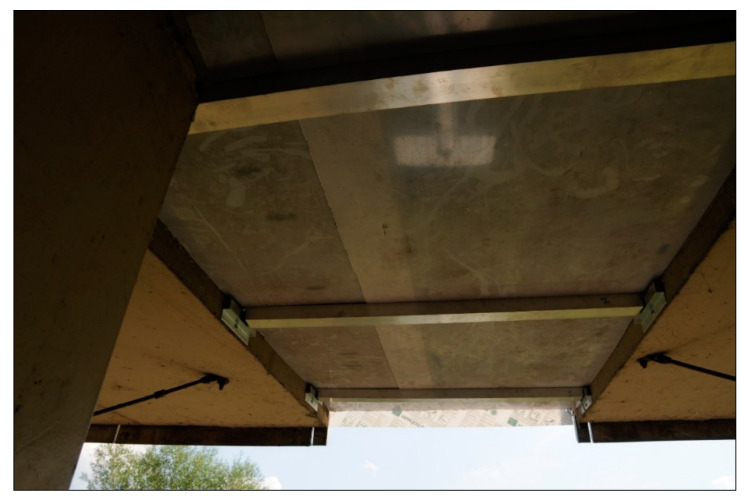
View of the solid polycarbonate plates mounted on the supporting structure of the acoustic cover.

**Figure 7 materials-13-02842-f007:**
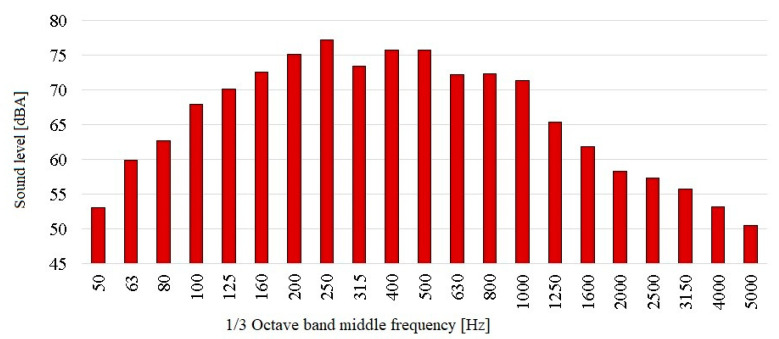
Results of measurements to determine the performance of the sound absorbing material inside the sound cover.

**Figure 8 materials-13-02842-f008:**
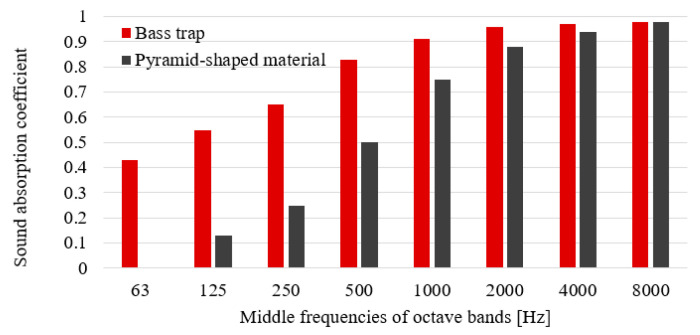
Sound absorption curves for both materials used.

**Figure 9 materials-13-02842-f009:**
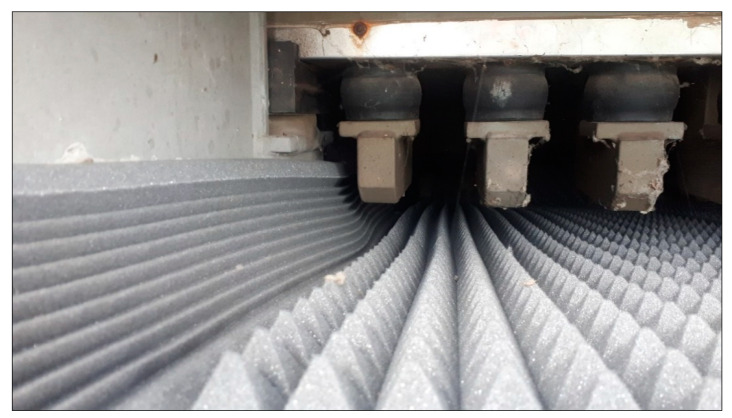
View of the sound absorbing materials from the inside of the sound cover (left side—the “bass trap”, right side—the “pyramid”.

**Figure 10 materials-13-02842-f010:**
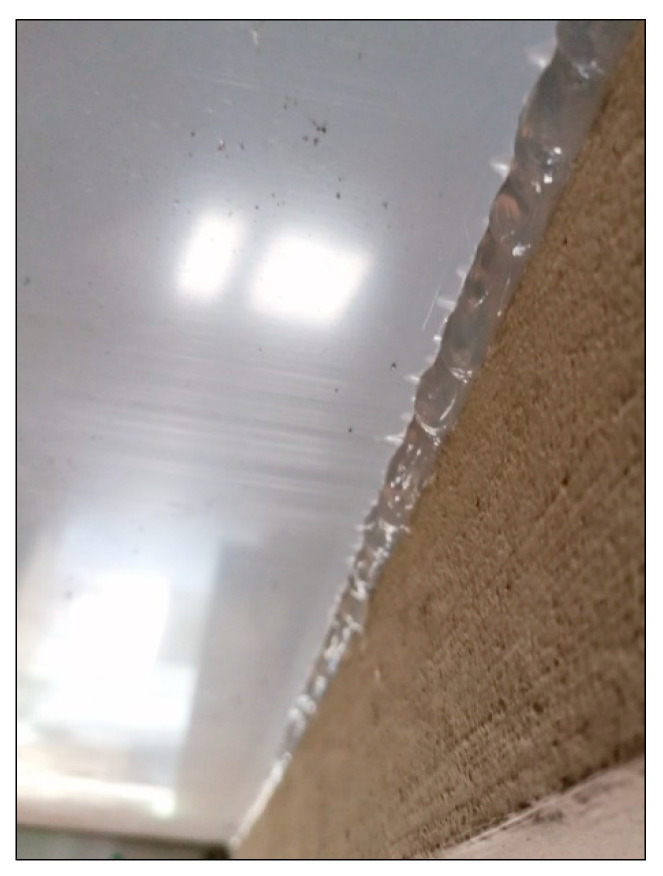
Sealing of the contact between the solid polycarbonate slab and concrete slab of the bridge span with silicone.

**Figure 11 materials-13-02842-f011:**
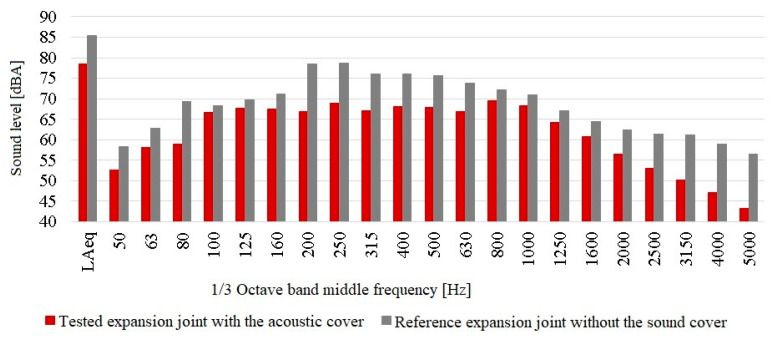
Results of the acoustic measurements at points located under the tested expansion joint equipped with the acoustic cover, and under the reference expansion joint.

**Figure 12 materials-13-02842-f012:**
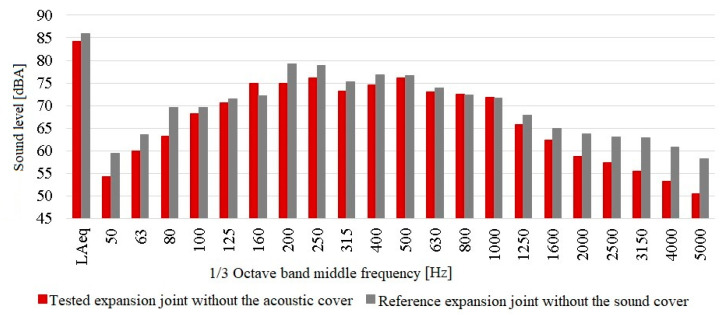
Results of the acoustic measurements at points located under the tested expansion joint without the acoustic cover, and under the reference expansion joint.

**Figure 13 materials-13-02842-f013:**
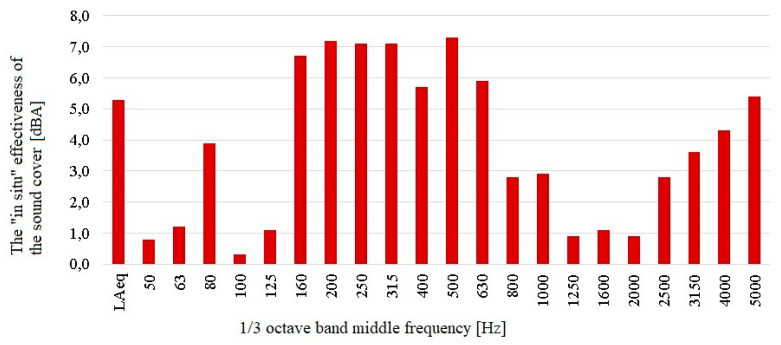
The “in situ” acoustic effectiveness of the prototype sound cover of the expansion joint device.

**Figure 14 materials-13-02842-f014:**
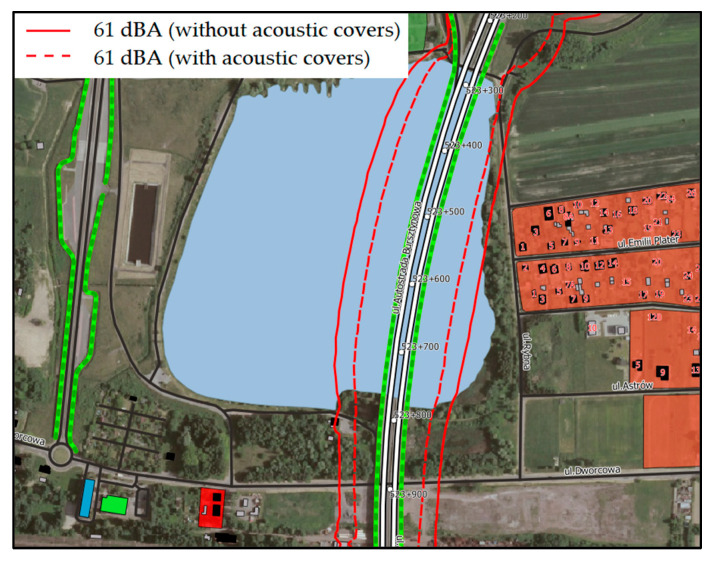
The results of the acoustic modeling—with and without the use of the acoustic covers on the expansion joints for a day.

**Figure 15 materials-13-02842-f015:**
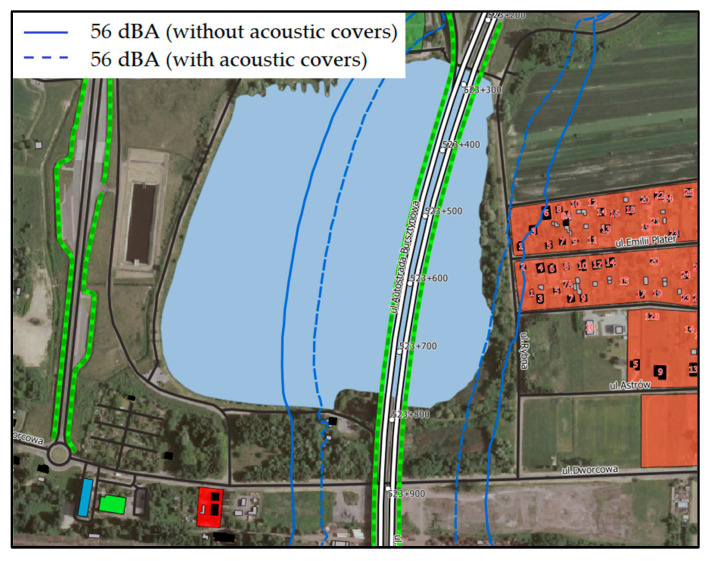
The results of the acoustic modeling—with and without the use of the acoustic covers on the expansion joints for a night.

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
