# Peer review of "Influence of the Acoustic Cover of the Modular Expansion Joint on the Acoustic Climate in the Bridge Structure Surroundings"

_materials, 2020, doi:10.3390/ma13122842_

Round 1

Reviewer 1 Report

attached you can find the document with comments and suggestions for Authors.

Author Response

Reviewer 1

  1. Unnecessary - there is only one corresponding author.

Ad. 1 Verse 9. Note was taken into account.

  1. This should be in the Acknowledgments section, not in the Abstract. According to the Instructions for Authors, Abstract should also include summarized main findings and main conclusions.

Ad.2 Verse 350-351. The information was saved in "Acknowledgments". Conclusions and findings were added to the abstract.

  1. A bridge expansion joint is not a gap! It is a structural component designed to provide smooth passage over the gap between adjacent sides of a bridge deck joint, while also permitting relative translation and/or rotation of the two sides of a bridge deck. It is also called "deck movement joint".

Ad.3. Verse 29-31. The authors improved the meaning of the word bridge expansion joint, confirming the correct definition of the reviewer.

  1. The problem occurs everywhere where there are vehicles traveling over bridges with expansion joints, not just in Europe, Japan, China and Australia.

Ad.4. Verse 40-42. The authors reworded the sentence regarding the occurrence of the analyzed problem.

  1. See reference [2]: "There are two components of noises or sounds; one is “joint sound”, which is radiated when moving vehicles pass over the expansion joints of bridges and another is “span sound”, which is radiated when vehicles run through the spans." Vehicles do not travel THROUGH objects.

Ad. 5. Verse 43.The authors reworded the sentence changing "through" to "over".

  1. „over”

Ad. 6. Verse 51. The authors improved the translation for "over".

  1. Why are these studies relevant for the paper?

Ad. 7 Verse 56-61. The modular expansion joints are one of the most commonly used expansion joints. From an acoustic point of view, however, they are one of the louder ones. One way to improve the acoustic climate within this type of expansion joint is to use steel overlays in the shape of e.g., rhombus. The object analyzed by the authors was also equipped with this type of expansion joints. The authors decided to devote one paragraph to the description of issues related to this type of dilatation in relation to the available literature.

  1. Why is this study relevant for the paper?

Ad.8. Verse 69-74. The authors wanted to emphasize the issues related to the impulse noise, which may affect the migration of wild animals at the animal crossings. Currently, due to the development of the linear infrastructure, which is directly related to the construction of new engineering facilities, more and more often the problem of impulse noise created at the junction of the wheel-bridge dilatation concerns a wider group not only of society, but also of the ecosystem.

  1. Vehicles are passing over, not through the expansion joint.

Ad. 9. Verse 93. The authors corrected the translation.

  1. What is "material sound"? Is it noise generated by vibrations of the elements?

Ad.10. Verse 96-97. The material sound is created by vibrations that occur when the car passes the expansion joint, in steel elements (main girders and cross-members) of the bridge. In other words, it is the sound induced in the bridge structure.

  1. Sound intensity near a hard surface is enhanced (because the reflected wave adds to the incident wave).

Ad.11. Verse 98-99. The comment was taken into account by the authors.

  1. It is unclear what is a "typical acoustic impact generated by passing vehicles" - impact from rolling noise and engine/transmission noise?

Ad.12. Verse 99-100. The authors confirm that by typical acoustic impact from passing vehicles they understand the rolling noise and engine noise.

  1. Noise generated by the vehicles passing over the joints is not "natural".

Ad.13. Verse 107. The authors took into account the reviewer's comment.

  1. The acoustic cover is a noise damper, not a noise barrier - it is attached to the structure that is transmitting vibrations (and therefore noise).

Ad. 14. Verse 109. The authors changed it to the "acoustic cover".

  1. A-weighted?

Ad. 15. Verse 115. The authors confirm the reviewer's comment. The measurement results were presented using an equivalent A-weighted sound level.

  1. Are these acoustic tests road traffic noise measurements (conducted to determine "typical acoustic impact generated by passing vehicles")?

Ad.16. Verse 158. The authors explain that the measurements were carried out to determine the reduction of noise caused by the acoustic cover in the vicinity of the object, and not to determine the level of the road noise. Due to the very high distortion, this was not possible, as the authors wrote in the paper.

  1. Describe input parameters for the road traffic noise model which were used in the calculation. What noise prediction software was used?

Ad.17. Verse 165-169. The note was taken into account by the authors.

  1. What method for industrial noise calculation was used? Which in-situ measurements were used to verify the model (only the measurements conducted under the expansion joints)? What was the noise source power?

Ad.18. Verse 173-176. The note was taken into account by the authors. For the acoustic calculations of industrial noise, the method described in the ISO 9613-2: 1996 „Acoustics — Attenuation of sound during propagation outdoors — Part 2: General method of calculation, was used.

  1. "Sound cover" is also used.

Ad.19. Verse 185. The authors have included the reviewer's comment.

  1. Rw is "weighted sound reduction index".

Ad.20. Verse 202. The authors have included the reviewer's comment.

  1. What kind of materials were used? What are their characteristics and sound absorption coefficients? On what part of the interior is each type used and why?

Ad.21. Verse 229-233. Inside the acoustic cover, materials were used, the description of which is given in the paper, and the view is shown in Figure 8. The authors extended the description, taking into account the reviewer's comment regarding the characteristics of the sound absorption coefficient.

  1. Use difference maps instead (only for day and night period).

Ad.22. Verse 309 and 311. The authors have included the reviewer's comment.

  1. Compare your results with those from similar studies. Future research directions should also be highlighted.

Ad.23. Verse 338-344. The results of the acoustic tests of the cover in question were not compared with other devices, because no literature results were found for solutions of the same type (sound-absorbing-insulating cover). In addition, the lack of a uniform testing methodology often prevents such comparisons. The reviewer's comment was taken into account in the directions of further research.

  1. https://doi.org/10.1006/jsvi.1996.0270

Ad.24. Verse 387. The authors added the DOI number to the reference No. 12 .

Reviewer 2 Report

I think this is a relatively well-prepared paper, but it is substantially out of the scope of the journal Materials. After reading the paper, I could not figure out why authors submitted here, they should have read better the aims and scope of the journal? It is of little interest for its readers...

There is no characterization whatsoever of the materials used for the "acoustic cover" as defined by the authors - no acoustical, chemical, mechanical (or physical in general) characterization of the absorbing modules.

This is an environmental acoustics/transportation noise study, not a materials one. I would recommend sending the paper to a more appropriate journal where authors will certainly be successful.

Author Response

Reviewer 2

I think this is a relatively well-prepared paper, but it is substantially out of the scope of the journal Materials. After reading the paper, I could not figure out why authors submitted here, they should have read better the aims and scope of the journal? It is of little interest for its readers..

There is no characterization whatsoever of the materials used for the "acoustic cover" as defined by the authors - no acoustical, chemical, mechanical (or physical in general) characterization of the absorbing modules.

This is an environmental acoustics/transportation noise study, not a materials one. I would recommend sending the paper to a more appropriate journal where authors will certainly be successful.

Reply:

The authors thank the reviewer very much for acknowledging that the paper is well prepared. We do not want to argue with the general comment of the reviewer that the paper is outside the scope of this journal. The authors have long wondered about the journal in which they could submit this article. It was a very difficult decision. The article is interdisciplinary and is related to research, acoustics, and the dilatation solution performed. Finally, we decided that the Materials will be the best choice because research and results relate to specific phenomena that are associated with the components (materials) used for the construction of the viaduct and expansion joints. Other dilatation solutions made of other materials are known. In this particular case, the goal was to show how such a complicated arrangement of the materials behaves due to the noise, i.e., the road-expansion joint-construction of the viaduct.

Reviewer 3 Report

The work appears interesting and the topics covered are suitable for publication. However, I would like to point out some points for improvement before publication.

The grammar and the construction of the sentences should be revised and in some cases checked in depth. Regarding the measurement methodology, it seems correct but should be explained in more detail. The sound absorbing material used (apparently acoustic foam) should be characterized and described more accurately, also presenting the absorption curves measured separately. In figure 7 it is not clear what is presented: the difference between the measurements with the cover, with and without the sound absorbing material? It should be better explained. About the absorbing material used, the choice made does needs some explanations, since foam has a certain absorption grade in the frequencies involved, but looking at how it was used and positioned (figure 8) perhaps it would have been interesting to investigate a material with a more marked insulation effect (e.g. glass wool or any other material with a higher mass) instead of absorption. Moreover, aspects of durability and sustainability of the materials used in the test are not covered. There is a lack of more detailed information on the simulations and their comparison with the experimental results described.

Author Response

Reviewer 3

The grammar and the construction of the sentences should be revised and in some cases checked in depth. Regarding the measurement methodology, it seems correct but should be explained in more detail. The sound absorbing material used (apparently acoustic foam) should be characterized and described more accurately, also presenting the absorption curves measured separately. In figure 7 it is not clear what is presented: the difference between the measurements with the cover, with and without the sound absorbing material? It should be better explained. About the absorbing material used, the choice made does needs some explanations, since foam has a certain absorption grade in the frequencies involved, but looking at how it was used and positioned (figure 8) perhaps it would have been interesting to investigate a material with a more marked insulation effect (e.g. glass wool or any other material with a higher mass) instead of absorption. Moreover, aspects of durability and sustainability of the materials used in the test are not covered. There is a lack of more detailed information on the simulations and their comparison with the experimental results described.

Reply:

The paper was revised regarding the language. The description of the research was specified in detail with the necessary parameters and properties of the materials used, and the parameters used in the theoretical analyzes (the model made in the SoundPlan) (Verse 165-169; 173-176; 228-233; 236). Figure 7 presents the results of measurements made directly under the tested expansion joint without the acoustic cover installed. They were to determine the frequency characteristics of sound emitted by expansion joints. On this basis, the parameters of sound-absorbing materials were selected in such a way that they absorb sound at dominant frequencies. The absorption curves for both materials used were supplemented in the article (verse 236). The authors included in the text information on the absorption coefficients for the materials used during the tests. Regarding other materials used to fill the interior of the cover, the authors agree with the reviewer and verification of the use of, e.g., the glass wool may be interesting from the point of view of the noise reduction. The authors had quite a limited time for the implementation of the project and resources, and in the future when conducting this type of research they will certainly apply this proposal. At the same time, the type of the material was discussed with the facility manager, who was very skeptical about other materials than those described in the paper. Nevertheless, the use of other material can be a further direction for the development of research and works on the use of acoustic cover. Manufacturers of acoustic materials used in the construction of the cover confirmed the possibility of using these materials in the conditions prevailing under the bridge structure. Based on this information, the possibility of their use in the examined place was assumed.

Author Response

Reviewer 5

  1. The usage of grammar and English are good. However, general review (Spelling check and sentence constructions in some places) will improve the paper further.

Ad.1. The language was revised by a native speaker.

  1. The sentence “The passage…steel elements” (line number 16-18) is not clear. It is recommended to rewrite for clarity.

Ad.2. Verse 16-18. The authors corrected the sentence.

  1. In line number 30, the word “prestression” needs to be corrected

Ad.3. Verse 30-31. As mentioned above, the language revision has been provided.

  1. In line number 156, the word “were not presented” needs to be corrected as “were presented”.

Ad.4. Verse 160. The authors explain that the measurements were carried out to determine the reduction of noise caused by the noise cover in the vicinity of the object, and not to determine the level of the road noise. Due to the very high distortion, this was not possible, as the authors wrote in the paper.

  1. Figures 5 and 8 were not referenced inside the text.

Ad.5. Verse 195-196 and 234-235. The authors supplemented the text with missing references to Fig. 5 and 8.

  1. It is recommended to change the heading of the section 5.

Ad.6. Verse 248. The authors rewritten the title of the section 5 to "Results of measurements"

  1. The data shown in Figure 7, 10, 11 and 12 are in dB scale or dBA scale?

Ad.7. Verse 223; 253; 271; 283. The authors agree with the reviewer. The description of the vertical axis scale in the graphs has been changed to the dBA.

Round 2

Reviewer 1 Report

Comments and suggestions:

"15. A-weighted?
Ad. 15. Verse 115. The authors confirm the reviewer's comment. The measurement results were presented using an equivalent A-weighted sound level." - indicate whether the measured and calculated noise levels are A-weighted (dB SPL is not equal to dBA) throughout the paper by adding the "A" to the "dB".

"22. Use difference maps instead (only for day and night period).
Ad.22. Verse 309 and 311. The authors have included the reviewer's comment." - the authors did not provide the requested noise map - the remark was to use "noise difference maps" instead of noise maps derived from acoustic modeling with and without the acoustic cover. These "difference maps" (one for "day" and one for "night" period) will give a clear overview of the decrease of noise. This can easily be done in SoundPLAN.

"23. Compare your results with those from similar studies. Future research directions should also be highlighted.
Ad.23. Verse 338-344. The results of the acoustic tests of the cover in question were not compared with other devices, because no literature results were found for solutions of the same type (sound-absorbing-insulating cover)." - compare the results of your study to the noise reduction possibilities of noise mitigation measures such as Concrete Enclosure of Joint Cavity, Helmholtz Resonator in Joint Cavity, and/or products like ROBO-MUTE Noise protection mats. These noise mitigation measures should also be mentioned in the Introduction.

Author Response

Reviewer 1

  1. "15. A-weighted?

Ad. 15. Verse 115. The authors confirm the reviewer's comment. The measurement results were presented using an equivalent A-weighted sound level." - indicate whether the measured and calculated noise levels are A-weighted (dB SPL is not equal to dBA) throughout the paper by adding the "A" to the "dB"..

Ad. 1 Verse: 22, 24, 153, 155, 157, 159, 161, 240, 271, 290, 301, 304, 314, 340, 342, 343. The authors took into account the reviewer's comment.

  1. Use difference maps instead (only for day and night period).

Ad.22. Verse 309 and 311. The authors have included the reviewer's comment." - the authors did not provide the requested noise map - the remark was to use "noise difference maps" instead of noise maps derived from acoustic modeling with and without the acoustic cover. These "difference maps" (one for "day" and one for "night" period) will give a clear overview of the decrease of noise. This can easily be done in SoundPLAN..

Ad.2 Verse 324 – 329. The authors took the reviewer's comment into account by changing the maps shown in Fig. 14 and Fig. 15.

  1. Compare your results with those from similar studies. Future research directions should also be highlighted.

Ad.23. Verse 338-344. The results of the acoustic tests of the cover in question were not compared with other devices, because no literature results were found for solutions of the same type (sound-absorbing-insulating cover)." - compare the results of your study to the noise reduction possibilities of noise mitigation measures such as Concrete Enclosure of Joint Cavity, Helmholtz Resonator in Joint Cavity, and/or products like ROBO-MUTE Noise protection mats. These noise mitigation measures should also be mentioned in the Introduction.".

Ad.3. Verse 78-96; 350-361 and 428-432. The authors took into account the reviewer's comment, adding in the introduction other methods (systems) for reducing the level of impulse noise and in the conclusions they referred to the comparison of obtained results from the in situ tests in relation to the results of tests from other systems available in the literature.

Reviewer 2 Report

Ok for me if Ok for Editor...

Author Response

Reviewer 2

Ok for me if Ok for Editor…

Reply:

The authors re-corrected the translation of the paper. They took into account the comments of other reviewers. Thank you for the positive opinion.
